# Malnutrition Screening Tools Are Not Sensitive Enough to Identify Older Hospital Patients with Malnutrition

**DOI:** 10.3390/nu15245126

**Published:** 2023-12-17

**Authors:** Carliene van Dronkelaar, Michael Tieland, Tommy Cederholm, Esmee M. Reijnierse, Peter J. M. Weijs, Hinke Kruizenga

**Affiliations:** 1Center of Expertise Urban Vitality, Faculty of Sports and Nutrition, Amsterdam University of Applied Sciences, 1067 SM Amsterdam, The Netherlands; m.tieland@hva.nl (M.T.); e.m.reijnierse@hva.nl (E.M.R.); p.j.m.weijs@hva.nl (P.J.M.W.); h.kruizenga@amsterdamumc.nl (H.K.); 2Department of Nutrition and Dietetics, Amsterdam UMC, Vrije Universiteit Amsterdam, 1081 HV Amsterdam, The Netherlands; 3Amsterdam Public Health, Aging & Later Life, 1081 HV Amsterdam, The Netherlands; 4Department of Public Health and Caring Sciences, Uppsala University, 751 22 Uppsala, Sweden; tommy.cederholm@pubcare.uu.se; 5Theme Ageing, Karolinska University Hospital, 14186 Stockholm, Sweden; 6Amsterdam Movement Sciences, Ageing & Vitality, 1081 HZ Amsterdam, The Netherlands

**Keywords:** undernutrition, older adults, diagnosis, GLIM, screening

## Abstract

This study evaluates the concurrent validity of five malnutrition screening tools to identify older hospitalized patients against the Global Leadership Initiative on Malnutrition (GLIM) diagnostic criteria as limited evidence is available. The screening tools Short Nutritional Assessment Questionnaire (SNAQ), Malnutrition Universal Screening Tool (MUST), Malnutrition Screening Tool (MST), Mini Nutritional Assessment—Short Form (MNA-SF), and the Patient-Generated Subjective Global Assessment—Short Form (PG-SGA-SF) with cut-offs for both malnutrition (conservative) and moderate malnutrition or risk of malnutrition (liberal) were used. The concurrent validity was determined by the sensitivity, specificity, positive predictive value (PPV), negative predictive value (NPV), and the level of agreement by Cohen’s kappa. In total, 356 patients were included in the analyses (median age 70 y (IQR 63–77); 54% male). The prevalence of malnutrition according to the GLIM criteria without prior screening was 42%. The conservative cut-offs showed a low-to-moderate sensitivity (32–68%) and moderate-to-high specificity (61–98%). The PPV and NPV ranged from 59 to 94% and 67–86%, respectively. The Cohen’s kappa showed poor agreement (k = 0.21–0.59). The liberal cut-offs displayed a moderate-to-high sensitivity (66–89%) and a low-to-high specificity (46–95%). The agreement was fair to good (k = 0.33–0.75). The currently used screening tools vary in their capacity to identify hospitalized older patients with malnutrition. The screening process in the GLIM framework requires further consideration.

## 1. Introduction

Malnutrition has a major impact on health outcomes and quality of life for hospitalized older patients [1,2]. Malnutrition is associated with a higher risk of complications, adverse functional outcomes, and increased mortality, independent of the underlying illness [3,4]. Hospitalization carries a high risk for the loss of muscle mass and function, which can be accelerated by malnutrition [5]. Early detection of malnutrition is therefore important in order to start nutritional treatment early during hospital admission and to continue it after discharge [1,6]. Currently, malnutrition is identified with various validated malnutrition screening tools [7]. Until recently, these screening tools have mainly been used to identify patients as either not at risk, at risk, or malnourished. In the Dutch context, the Short Nutritional Assessment Questionnaire (SNAQ) is established for identifying patients with malnutrition for dietetic assessment and treatment [8,9]. In 2019, a new framework for diagnosing malnutrition was established by the Global Leadership Initiative on Malnutrition (GLIM), using a two-step approach [10]. In the first step, the use of a screening tool is suggested to screen for any patient potentially at risk of malnutrition. The second step is a confirmation of malnutrition based on a set of five criteria: three phenotypic criteria, i.e., unintentional weight loss, low BMI, and low muscle mass, and two etiologic criteria, i.e., reduced food intake or assimilation and disease burden determined by inflammation. Malnutrition is then diagnosed if at least one phenotypic and one etiologic criterion is met. The GLIM criteria have been shown to have a high diagnostic accuracy for identifying patients with malnutrition, with a sensitivity of 81% and a specificity of 80% [4,11]. Despite the high predictive ability of the GLIM malnutrition diagnosis for complications, length of hospital stay, and mortality in older adults, the feasibility of applying the criteria in different clinical care settings remains challenging [12,13].

A recent scoping review, analyzing mainly retrospective cohort studies, showed that only a third of the studies applied the full two-step approach; only 52% of the studies applied all five criteria, and up to 42% of the studies did not clearly describe the methods used to apply the GLIM criteria [14]. Within the first step of the GLIM framework, no specific recommendations are made on which screening tool should be used in which population or setting [10]. In the scoping review, it was highlighted that there is a need to evaluate the screening tools used in the first step of the criteria as these tools were developed and validated against various previous sets of criteria for malnutrition [14].

This study aimed to evaluate the concurrent validity of five screening tools for malnutrition: the Short Nutritional Assessment Questionnaire (SNAQ), the Malnutrition Universal Screening Tool (MUST), the Malnutrition Screening Tool (MST), the Mini Nutritional Assessment—Short Form (MNA-SF), and the Patient-Generated Subjective Global Assessment—Short Form (PG-SGA-SF) with respect to identifying older hospitalized patients with potential malnutrition according to the GLIM criteria.

## 2. Materials and Methods

### 2.1. Study Design and Participants

Data were collected from January 2021 to December 2022 in five hospitals in Amsterdam, the Netherlands: the university hospital Amsterdam University Medical Center (Amsterdam UMC) at the Academic Medical Center (AMC) (~1000 beds) and VU University Medical Center (VUmc) (~730 beds) locations; the teaching hospital Onze Lieve Vrouwen Gasthuis (OLVG), at the East (555 beds) and West (365 beds) locations; and the regional hospital BovenIJ (315 beds). Patients from several wards were assessed for eligibility: the acute admission ward, internal medicine, cardiology, gastroenterology, neurology, and the geriatric ward. With the BovenIJ hospital being a smaller regional hospital, several specialties there were grouped in one ward. In that case, all eligible patients regardless of specialized medical need were included. Patients were eligible when aged 55 or over, admitted to the hospital in the last 48 h, and having a reasonable understanding of the Dutch language. Patients were excluded if they were being nursed in contact isolation due to COVID-19 or other infectious diseases, suffered from severe cognitive impairment or delirium, or had end-of-life palliative care based on the judgment of the attending nurse. From all included patients, written informed consent was obtained. All questionnaires and measurements were performed by trained research staff. The medical ethical committee of the Amsterdam UMC, VUmc (2019.680) location, approved the study, and the study followed the Declaration of Helsinki.

### 2.2. Screening Tools

Five screening tools were used to screen for malnutrition, namely, the SNAQ, MUST, MST, MNA-SF, and PG-SGA-SF, to conduct a comprehensive evaluation of the most often used screening tools for older patients in a hospital setting within the Netherlands, based on the outcomes of a meta-analysis by Power et al. [7]. The SNAQ consists of questions on unintentional weight loss in the past month, decreased appetite, and the use of oral nutritional support that had the best ability to predict malnutrition [8]. The questionnaire was developed for hospitalized patients above 18 years of age. The MUST considers low BMI and unintentional weight loss over the past three to six months and asks if a patient has been ill and unable to eat for more than five days [15]. The MST evaluates adults in various settings and consists of questions on unintentional weight loss over the last six months and decreased appetite; it was developed for hospitalized patients 18 years or older [16]. The MNA-SF is the longest questionnaire of the five screening tools and is designed for older adults. The MNA-SF takes more risk factors into account and consists of questions on weight loss over the last three months, decreased intake, mobility, psychological stress or acute illness in the past three months, neuropsychological problems, and BMI [17]. The PG-SGA was developed for adult cancer patients and considers multiple risk factors. The PG-SGA-SF (box 1 to 4) classifies weight loss over the last month if available, otherwise, weight loss over the past six months. Questions on food intake and physical activity over the past month and problems leading to decreased food intake over the past two weeks are part of the PG-SGA-SF [18]. The PG-SGA-SF was added as the fifth screening tool in August 2022 and from that time point onwards applied to every patient included in this study.

The screening tools use different cut-offs to identify patients as well-nourished, malnourished, moderately malnourished, or at risk of malnutrition. In this study, patients were classified as malnourished based on the screening tools, without confirmation based on the GLIM criteria, if they scored ≥3 points on the SNAQ (max. 7 points), ≥2 on the MUST (max. 6 points) or MST (max. 5 points), ≤7 points on the MNA-SF (max. 14 points), and ≥9 points on the PG-SGA-SF (max. 35 points) [8,15,16,17,18]. These cut-offs are considered as conservative cut-offs. The SNAQ, MUST, MNA-SF, and PG-SGA-SF also have more liberal cut-offs available to identify moderate malnutrition or risk of malnutrition. Additional analyses were conducted with these more liberal cut-offs (≥2 points on SNAQ, ≥1 point on MUST, ≤11 points on MNA-SF, and ≥4 points on the PG-SGA-SF). Finally, as the MNA-SF is specially developed for older patients, additional analyses were conducted comparing patients aged < 70 years with those aged ≥ 70 years.

### 2.3. GLIM Criteria

The GLIM definition was used to identify malnutrition. Questions to assess unintentional weight loss, weight at admission, and weight from 1 month, 3 months, 6 months, and 12 months before admission were asked of the patient. In case the patient did not remember their weight, the electronic medical record was checked for available information on weight; if unavailable, weight was considered missing and excluded from analyses. Height and weight at admission were used to calculate the BMI. Bioelectrical impedance analysis (BIA) measurements were performed to assess muscle mass. The BIA was performed with a hand-to-foot device (BodyStat 500 or Quadscan 4000; BodyStat Body Composition Technology, Cronkbourne, UK), with the patient in a supine position with four electrodes connected to one side of the body and arms not touching the trunk and legs slightly separated. BIA measurements were performed between 9 a.m. and 1 p.m. Patients were not in a fasted state before the measurement [19,20]. BIA measurements were not performed if the patient had severe edema or a pacemaker, was wearing a heart monitor, or had an IV drip in both hands or arms, according to hospital protocols [21]. The impedance, reactance, and resistance at 50 kHz of the BIA were used to apply the Rutten [22], Sergi [23], and Janssen [24] equations to determine the fat-free mass index (FFMI), the appendicular skeletal muscle mass index (ASMI), and the skeletal muscle mass index (SMI), respectively. Within the second step of the GLIM framework, cut-offs for low muscle mass are recommended for FFMI as well as ASMI and SMI. For the main analysis, the cut-off for FFMI was used to determine low muscle mass, as the BIA was performed on the whole body and is used in that manner within the Dutch guidelines for identifying malnutrition [25]. The other recommended measures of muscle mass (i.e., ASMI and SMI) were used to compare the impact of the selected measure on the prevalence of malnutrition. To assess the criterion of reduced intake and/or malabsorption, the question of the MNA-SF on reduced intake over the past three months and a question on malabsorption, stating ‘Did you suffer from dysphagia, nausea, vomiting, diarrhea, constipation, or abdominal pain?’, were used. Inflammation was defined using CRP or (pre-) albumin levels from three days before inclusion, and data were obtained from patients’ medical files when available.

Patients were classified as malnourished based on the GLIM criteria, without prior screening, when at least one phenotypic and one etiologic criterion were fulfilled.

The phenotypic criteria were applied as follows:Weight loss: >5% within the past 6 months or >10% within the past 12 months [10];Low BMI: <20 kg/m^2^ for patients under 70 years old; <22 kg/m^2^ when aged 70 years or older [10];Reduced muscle mass: males with an FFMI of <17 kg/m^2^ and females with an FFMI of <15 kg/m^2^ [10];

The etiologic criteria were applied as follows:


Reduced food intake or assimilation: having a severely decreased appetite or having answered ‘yes’ on the malabsorption question, ‘Did you suffer from dysphagia, nausea, vomiting, diarrhea, constipation, or abdominal pain?’ [10];Inflammation or disease burden: having CRP levels of ≥3 mg/L or pre-albumin levels of <30 mg/dl or albumin levels of <3.8 g/L [26,27].


### 2.4. Statistical Analysis

Patient characteristics were analyzed with descriptive statistics and presented as means and standard deviations, medians, and interquartile ranges or as frequencies and percentages. Sensitivity (the capacity to identify true positive cases), specificity (the capacity to identify true negative cases), positive predictive value (PPV), and negative predictive value (NPV) were calculated, and the Cohen’s kappa coefficient (indicating agreement) was determined for each screening tool against the GLIM criteria. Sensitivity and specificity were considered low if <50%, moderate if 50–80%, and high if >80% [7,28]. Cohen’s kappa measures the agreement between the two tools and was classified as follows: <0.20 poor, 0.20–0.40 fair, 0.40–0.60 moderate, 0.60–0.80 good, and >0.80 very good [29,30]. The McNemar test was performed to evaluate if the screening tools identified the same patients as the GLIM criteria. A non-statistically significant McNemar test would indicate that the same patients are being identified. Univariate logistic regression was performed to identify the contribution of different criteria and cut-offs of the screening tools on the GLIM criteria. Subjects with incomplete basic data (i.e., missing data on either age, sex, weight, or height), or those who were included in the study within 30 days from the previous inclusion, were omitted from the analyses. In addition, subjects for whom one of the five GLIM criteria could not be assessed due to missing data were excluded from the analyses. Statistical significance was set at α < 0.05. All analyses were performed in SPSS for Windows version 28.0.

## 3. Results

### 3.1. Patient Characteristics

A total of 2623 patients gave informed consent and were screened for malnutrition. Of these, 55 patients (2%) had incomplete basic data (i.e., age, sex, height, or weight) and ten (0.4%) were included within 30 days since their prior inclusion and therefore excluded from the analyses. In addition, we were only able to perform a BIA measurement on 430 patients. The final analyses were performed on patients with complete data on all five criteria of the GLIM, leading to a sample size of 356 patients (Figure 1).

The included patients had a median age of 70 years (interquartile range 63–77 years) (range 55–98 years), and 54% were male (Table 1). The study population was a heterogenous group of older patients as 72% of them were screened at the acute admission ward, which has a large variety of medical specialties.

### 3.2. Prevalence of Malnutrition

Of the patients screened after August 2022, 126 patients had complete GLIM data and had therefore data on the PG-SGA-SF. The prevalence of malnutrition according to the different screening tools, without confirmation by the GLIM, ranged from 18 to 52% (Table 2). The prevalence according to the GLIM criteria was 42% when no screening tool was used a priori (Table 3). When the GLIM criteria were analyzed separately, the criteria with the highest proportions were unintended weight loss (32%), reduced intake (71%), and inflammation (83%). When the SMI measure (22%) was used, the prevalence of low muscle mass was similar to the FFMI measure, but the prevalence was higher when the ASMI measure was used (45%). Using these measures resulted in a prevalence of malnutrition based on the GLIM criteria of 44% and 54% for the SMI and ASMI measures, respectively.

### 3.3. Concurrent Validity

When the conservative cut-offs of the screening tools were used to identify malnutrition, they performed with low-to-moderate sensitivity (32–68%) and moderate-to-high (61–98%) specificity (Table 4) in relation to the GLIM. Cohen’s kappa showed that the agreement between each screening tool and the GLIM criteria was poor to moderate (0.21–0.59). The McNemar test showed that the screening tools identified different patients as malnourished in comparison to the GLIM criteria (*p* < 0.001). Only for the PG-SGA-SF was this was not the case (*p* = 0.233). When the liberal cut-off points, i.e., for identification of moderate malnutrition or risk of malnutrition, were used, the corresponding sensitivity was moderate to high (66–89%) and the specificity low to high (46–95%). The agreement based on Cohen’s kappa was fair to good (0.33–0.75). The highest sensitivity was observed for PG-SGA-SF and MNA-SF, 89% and 86%, respectively.

Comparing patients aged < 70 years to those aged ≥ 70 years, only the MNA-SF showed a slightly higher sensitivity when the liberal cut-off was used (82% and 89%, respectively). For the other screening tools and cut-off points, the sensitivity was higher with an average of 15% for those aged < 70 years (Appendix A).

The effect of applying different measures of low muscle mass within the GLIM criteria on the sensitivity and specificity of the screening tools can be found in Appendix A. The SMI measure for low muscle mass showed similar results as when the FFMI measure for low muscle mass was used within the GLIM criteria. However, if the ASMI measure for low muscle mass was used within the GLIM criteria, the sensitivity of the screening tools dropped by ~10%, with the SNAQ screening tool shifting from moderate to low. Only in the PG-SGA did the concurrent validity remain similar to the FFMI measure.

## 4. Discussion

The GLIM framework advises the use of any validated screening tool in the first step to identify patients at risk of malnutrition before applying the phenotypic and etiologic criteria to diagnose malnutrition. This observational study showed that the currently frequently used screening tools in hospital settings (SNAQ, MUST, MST, MNA-SF, and PG-SGA-SF) were unable to identify 32–68% of the hospitalized older adults with malnutrition according to the GLIM criteria when applying the conservative cut-offs for malnutrition. When the more liberal cut-offs, i.e., for moderate malnutrition or risk of malnutrition, were used, the sensitivity was higher, especially for PG-SGA-SF and MNA-SF. The concurrent validity of the screening tools varied greatly depending on the measure for low muscle mass used and whether applied to those aged < 70 years compared to those ≥70 years.

Similar to our findings, a Brazilian study in older hospitalized patients showed that when the MNA-SF (liberal cut-off) was used in the first step of the GLIM framework, 83% of the older patients in the emergency ward were identified as at risk of malnutrition, of which 50% were confirmed as malnourished with the GLIM criteria [31]. Another Brazilian study compared the liberal cut-offs of the MST, NRE-2017, NSR-2002, and SNAQ to the GLIM criteria in a general hospital population [32]. Although their population was younger (mean age 56 y), they found similar concurrent validity outcomes for the MST and the SNAQ as we found in our study. In a study with an older patient population (mean age 78 y), when the liberal cut-off was used, the MUST had lower concurrent validity than in our study, confirming that the MUST screening tool might be less appropriate for older hospital patients [33]. In line with our findings, two studies showed a low-to-fair agreement between the screening tools PG-SGA-SF and MST against the GLIM criteria, where the GLIM criteria had a better predictive ability for mortality [28,34].

Our study therefore adds to the body of work that suggests that the first step of the GLIM framework needs to be further analyzed and more thoroughly considered. Although the PPVs of the screening tools were high, meaning that of those identified as malnourished by the screening tool most were also identified as malnourished by the GLIM criteria, the low sensitivity indicated that half of the patients with malnutrition were not identified by the screening tools. In a clinical setting, a screening tool needs to have a high sensitivity, to be able to start treatment for those at risk. Furthermore, we showed that the choice of screening tool and cut-off points has major implications for those who will be subject to further nutritional assessment and potential nutrition treatment. For example, currently, over 80% of Dutch hospitals use the SNAQ screening tool with the conservative cut-off to identify potential patients with malnutrition [9]. Only for patients with a SNAQ malnutrition score (3 or higher) is a dietitian consulted to further assess nutritional status. Based on the findings of our study, this would mean that 34–54% of patients with malnutrition, based on GLIM criteria, are not assessed by a dietitian and do not receive nutritional treatment. Thus, for the implementation of the GLIM framework, a more sensitive screening procedure is warranted since specificity is provided by the GLIM confirmatory diagnostic step.

The prevalence of malnutrition based on the GLIM criteria was higher than expected in our study population [32,33,34,35]. Despite the high prevalence of malnutrition in older hospitalized patients, nurses and medical staff experience difficulties in diagnosing malnutrition due to a lack of knowledge and skills [36]. Early diagnosis and treatment are essential to prevent negative health outcomes [1]. Hence, identifying malnutrition in the hospital setting should be quick and easy. A decade ago, medical records were only paper-based. In the meantime, technology within the hospital has developed tremendously, and electronic medical records have become available in most hospitals. This creates the opportunity for systems and algorithms in electronic patient files, allowing a more automated screening for malnutrition and the risk of malnutrition. The GLIM criteria of unintended weight loss, low BMI, reduced intake, and inflammation could be built into these electronic patient records. For example, when current height and weight and weight from six and twelve months ago are added to the record, unintended weight loss and low BMI can be calculated automatically. Taking this approach could improve the screening process. By adding setting-specific malnutrition risk factors, e.g., cognition, mobility, and marital status, the sensitivity of the screening procedure could be further ameliorated as the first step of the GLIM framework. In addition, this automated process could potentially ease the workload for nurses. The diagnosis of malnutrition could then be completed by the measurement of muscle mass, as a complete assessment of all five criteria is essential to be certain that malnutrition is not present. Further studies are needed to validate this approach and assess its applicability. When electronic medical records are not present, MNA-SF with a cut-off of ≤11 points could be used in the first step of the GLIM framework, especially for older patients. The second step of the GLIM framework, i.e., assessment and diagnostics, could then be completed by a dietitian.

Another important finding is the variability in malnutrition prevalence rates depending on the chosen measure for low muscle mass, a factor that is recommended by the GLIM consensus. Although BIA has been validated for the assessment of muscle mass, it has its limitations. BIA estimates muscle mass by relying on measurements of total body water and equations for estimating fat-free mass (FFM) or skeletal muscle mass (SMM). Changes in FFM tend to reflect changes in muscle mass over a longer period [37]. Another approach is to assess appendicular skeletal mass (ASM), which is the sum of the lean soft tissue in the arms and legs. ASM is more sensitive to muscle mass changes and could be a more appropriate measure for evaluating nutritional status. However, when using BIA on a whole-body level, a segmental approach (ASM) relies on more assumptions than a whole-body approach (FFM). To use the ASM measure for the GLIM criteria, a more valid measure like DXA or CT would be preferable. A recent guidance paper for the assessment of low muscle mass within the GLIM still recommends both FFMI and ASMI measures for low muscle mass, but our study showed that these measures have an impact on the prevalence of malnutrition in older hospitalized patients [38]. This indicates that the recommended measures of low muscle mass require evaluation in the GLIM criteria.

One of the strengths of this study is that we had access to the data of 2623 patients, of which we could filter a complete dataset on the GLIM criteria of over 300 patients. Because of this, we were able to include a heterogenous patient population from several hospitals and admission wards, which makes our results not disease- or hospital-specific but more generalizable to other older patient populations. However, in this heterogenous patient population, it was more difficult to assess low muscle mass with the BIA due to the presence of heart monitors, pacemakers, ICDs, or IV drips in both hands/arms. The missing data on low muscle mass could have led to underreporting of the prevalence of malnutrition. Still, this also reflects the ‘real’ clinical setting and shows the feasibility of applying the GLIM criteria, including the muscle mass measurement with a BIA, in a clinical setting. A major methodological strength of our study is the simultaneous screening and assessment of the GLIM criteria. This avoids any day-to-day variance within patients concerning the measurements. Although we were able to collect data on the admission ward, we did not have access to data on the reason for admission. With this information, subgroup analyses could have been performed to assess if the prevalence of malnutrition and the concurrent validity of the screening tools were different among different reasons for admission.

## 5. Conclusions

The results of this study indicate that the currently used screening tools for malnutrition in hospital settings are not sensitive enough to identify older patients who are malnourished according to the GLIM criteria. The first step of the GLIM framework therefore requires further consideration. Using electronic medical records to screen for malnutrition might be an option for improvement, where information on the GLIM criteria of unintended weight loss, low BMI, reduced intake, and inflammation can be easily assessed. Further studies are needed to validate this approach and assess its feasibility. When electronic medical records are absent, the MNA-SF with the liberal cut-off (≤11 points) could be used for the screening step of the GLIM diagnostic procedure.

## Figures and Tables

**Figure 1 nutrients-15-05126-f001:**
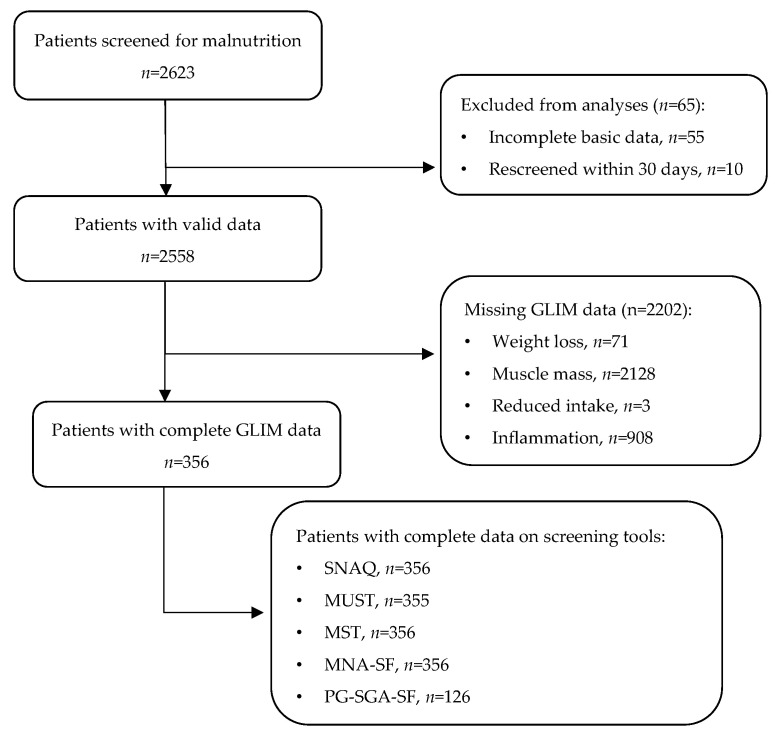
Flowchart of selection of final analysis sample. Several patients had more than one missing data point in their GLIM data.

**Table 1 nutrients-15-05126-t001:** Study subject characteristics.

Patients	*n* = 356
Sex, males, *n* (%)	192 (54)
Age in years, median (IQR)	70 (63–77)
BMI in kg/m^2^, median (IQR)	24.8 (22.6–28.1)
Hospital, *n* (%)	
Amsterdam UMC, location AMC	22 (6)
Amsterdam UMC, location VUmc	152 (43)
OLVG, location East	110 (31)
OLVG, location West	43 (12)
BovenIJ Hospital	29 (8)
Ward, *n* (%)	
Acute admission	257 (72)
Internal medicine	18 (5)
Cardiology	27 (8)
Neurology	18 (5)
Pulmonary	13 (4)
Gastroenterology	10 (3)
Geriatric	3 (<1)
Other	14 (4)

**Table 2 nutrients-15-05126-t002:** Prevalence of malnutrition and moderate malnutrition or risk of malnutrition according to the screening tools, without the confirmation diagnostic step of GLIM.

	*n*	Prevalence, *n* (%)
SNAQ	356	
Malnutrition (≥3)		88 (25)
Moderate malnutrition (≥2)		115 (32)
MUST	355	
Malnutrition (≥2)		65 (18)
Risk of malnutrition (≥1)		126 (36)
MST	356	
Malnutrition (≥2)		111 (31)
MNA-SF	356	
Malnutrition (≤7)		52 (15)
Risk of malnutrition (≤11)		206 (60)
PGSGA-SF	126	
Malnutrition (≥9)		65 (52)
Risk of malnutrition (≥4)		88 (70)

SNAQ: Short Nutritional Assessment Questionnaire; MUST: Malnutrition Universal Screening Tool; MST: Malnutrition Screening Tool; MNA-SF: Mini Nutritional Assessment—Short Form; PG-SGA-SF: Patient-Generated Subjective Global Assessment—Short Form.

**Table 3 nutrients-15-05126-t003:** Prevalence of malnutrition according to the GLIM criteria, without prior screening, and the occurrence of each criterion.

	*n*	Prevalence, *n* (%)
GLIM	356	148 (42)
*Phenotypic criteria*	356	156 (44)
Weight loss	356	113 (32)
Low BMI	356	59 (17)
Low muscle mass	356	83 (23)
*Etiologic criteria*	356	330 (93)
Reduced intake	356	251 (71)
Inflammation	356	294 (83)

GLIM: Global Leadership Initiative on Malnutrition.

**Table 4 nutrients-15-05126-t004:** Concurrent validity of the SNAQ, MUST, MST, MNA-SF, and PG-SGA-SF on malnutrition (A) and moderate malnutrition or risk of malnutrition (B) against the GLIM criteria.

	SNAQ (*n* = 356)		MUST (*n* = 356)		MST (*n* = 356)	MNA-SF (*n* = 356)		PG-SGA-SF (*n* = 126)	
	A (≥3)	B (≥2)	A (≥2)	B (≥1)	A (≥2)	A (≤7)	B (≤11)	A (≥9)	B (≥4)
False positive, *n*	5	17	7	11	16	5	79	27	38
False negative, *n*	65	50	89	32	53	203	21	18	6
Sensitivity, %	56	66	40	78	64	32	86	68	89
Specificity, %	98	92	97	95	92	98	62	61	46
PPV, %	94	85	90	91	86	90	62	59	57
NPV, %	76	79	70	86	78	67	86	71	84
Cohen’s kappa	0.57	0.60	0.39	0.75	0.59	0.21	0.45	0.29	0.33
McNemar	*p* < 0.001	*p* < 0.001	*p* < 0.001	*p* = 0.002	*p* < 0.001	*p* < 0.001	*p* < 0.001	*p* = 0.233	*p* < 0.001

A: malnutrition (cut-off point); B: risk of/moderate malnutrition (cut-off point); PPV: Positive Predictive Value; NPV: Negative Predictive value; SNAQ: Short Nutritional Assessment Questionnaire; MUST: Malnutrition Universal Screening Tool; MST: Malnutrition Screening Tool; MNA-SF: Mini Nutritional Assessment—Short Form; PG-SGA-SF: Patient-Generated Subjective Global Assessment—Short Form; GLIM: Global Leadership Initiative on Malnutrition.

## Data Availability

The data presented in this study are available on request from the corresponding author. The data are not publicly available due to privacy restrictions.

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
