# Peer review of "Malnutrition Screening Tools Are Not Sensitive Enough to Identify Older Hospital Patients with Malnutrition"

_nutrients, 2023, doi:10.3390/nu15245126_

Round 1

Reviewer 1 Report

Comments and Suggestions for Authors

I just have a few questions about the survey methodology and the collection of the research material:

1. in section 2.1: What about patients with exacerbations of chronic diseases? Were they excluded/included in the study? Why was the inclusion criterion set at age over 55? The MNA®-SF is a screening tool to help identify elderly patients who are malnourished or at risk of malnutrition. Who performed the distribution and collection of the study questionnaires. Did patients complete the questionnaires on their own? Who took patient measurements?

2. L165: Section 2.4. should begin with a capital letter.

3. Some of the references should be replaced with more up-to-date ones.

Reviewer 2 Report

Comments and Suggestions for Authors

The paper is interesting and well-written.

My major concern is the low proportion of patients with a complete dataset (300/2600), i.e. less than 15% of the original sample.

Another concern is relative to the definition of disease burden OR inflammation by either CRP levels ≥3 mg/L OR low pre-albumin or albumin levels. This criterion can therefore be satisfied by reduced albumin values which could depend  by a liver or kidney disease rather than by malnutrition.

bb

Reviewer 3 Report

Comments and Suggestions for Authors

The Authors decided to assess the validity of different screening tools for malnutrition detection. As the authors mentioned, it is the important topic, as undernutrition is the factor of unfavorable outcomes in patients, including elderly. 

Nonetheless I would like to express my concerns about the manuscript.

Please explain the abbreviations used in the abstract.

Why NRS 2002 or WHO (BMI classification) criteria were not included?

Why the inclusion criteria were set at the age of 55 years old, not 65 or 70?

If BMI was performed in the course of the study, I suggest adding the raw BIA results to the analysis and discussion.

Please provide more details of BIA measurements- time of day, special requirements or contradictions.

Bland Altman analysis is the tool dedicated for validation purposes. Please perform BA analysis.

Line 337 - the authors mention the limitations of BIA applications in cardiovascular patients with pacemakers or ICDs. Please provide relevant citation.

Please correct the typos in the text.

Comments on the Quality of English Language

Few typos.

Round 2

Reviewer 3 Report

Comments and Suggestions for Authors

Thank you very much for your corrections. The manuscript was highly improved. The authors replied to my comments.

However, there are still some typos - please correct them.

I also still suggest adding more relevant citations to the study rationale and methodology. 

Comments on the Quality of English Language

Please correct the typos.

Author Response

Thank you for your time revising the rebuttal. 

The program Grammarly was now used to check the manuscript for typos and improve the readability of the manuscript. Changes are made with tracked changes function in word.

The references of introduction and methods were checked for relevance and several references are added. Changed references are highlighted in yellow in the manuscript.